# Ground-to-UAV, laser-based, emissions quantification of methane and acetylene at long standoff distances

Kevin C. Cossel[1*], Eleanor M. Waxman[1], Eli Hoenig[1], Daniel Hesselius[2], Christopher Chaote[2], Ian Coddington[1], Nathan R. Newbury[1]

[1]National Institute of Standards and Technology, Spectrum Technology and Research Division, Boulder, CO 80305, USA
[2]Integrated Remote and In-Situ Sensing (IRISS), University of Colorado, Boulder, CO 80305, USA

*Correspondence to*: Kevin C. Cossel (kevin.cossel@nist.gov)

**Abstract.** Determination of trace gas emissions from sources is critical for understanding and regulating air quality and climate change. Here, we demonstrate a method for rapid quantification of the emission rate of multiple gases from simple and complex sources using a mass-balance approach with a spatially scannable open-path sensor – in this case, an open-path dual-comb spectrometer. The open-path spectrometer measures the total column density of gases between the spectrometer and a retroreflector mounted on an unmanned aerial vehicle (UAV). By measuring slant columns at multiple UAV altitudes downwind of a source (or sink), the total emission rate can be rapidly determined without the need for an atmospheric dispersion model. Here, we demonstrate this technique using controlled releases of $CH_4$ and $C_2H_2$. We show an emission rate determination to within 56% of the known flux with a single 10-minute flight and within 15% of the known flux after 12 flights. Furthermore, we estimate a detection limit for $CH_4$ emissions to be 0.03 g $CH_4$/s. This detection limit is approximately the same as the emissions from 25 head of beef cattle and is less than the average emissions from a small oil field pneumatic controller. Other gases including $CO_2$, $NH_3$, HDO, ethane, formaldehyde (HCHO), CO, and $N_2O$ can be measured by simply changing the dual-comb spectrometer.

## 1 Introduction

Measurements of the emission rate of a gas or gases from point and area sources are important in a range of monitoring applications. Several examples include measurement of emissions of $CH_4$ and volatile organic compounds (VOCs) from oil and gas facilities (well pads, compressor stations, processing plants, etc.), from landfills and from composting facilities, $CH_4$ and $N_2O$ (as well as VOCs) from waste-water treatment plants, $CH_4$, $NH_3$, and $N_2O$ from agricultural sites, and VOCs from industrial facilities. In all these examples, there are several important challenges for a measurement system. First, it is desirable to be able to measure emissions of multiple gases simultaneously. Second, the measurement system should be able to handle complex sources such as distributed sources or collocated sources. Finally, it is often desirable to be able to rapidly survey different sources to determine if emissions are present and then to quantify the emissions.

Because of the fundamental importance of emission rate measurements in monitoring and regulation, there are a wide range of different measurement methods that have been developed, all with distinct advantages and disadvantages. We cannot exhaustively review all the techniques here, but instead highlight a few general classes of techniques. First are the ground-based survey techniques with a point sensor (Vaughn et al., 2017; Zhang et al., 2019; Ravikumar et al., 2019; Riddick et al., 2022). One limitation of these techniques is that the vertical distribution of the gas is not measured, so assumptions need to be made about the vertical and horizontal gas dispersion (using an atmospheric transport model). The transport assumptions can be removed by releasing a tracer gas co-located with the unknown emission source (Czepiel et al., 1996; Mønster et al., 2014; Roscioli et al., 2015); however, this requires access to and knowledge of the emission source. To circumvent these challenges, mass-balance approaches can be used by measuring in two dimensions which greatly relaxes the requirements on the transport model. Most frequently, mass-balance is performed using point sensors on aircraft (White et al., 1976; Alfieri et al., 2010; Karion et al., 2013; Conley et al., 2017). While effective for large sources, challenges such as flight altitude restrictions, cost, and the requirement for fast sensors can limit applicability and repeat measurements. More recently, unmanned aerial vehicles (UAVs) have been used for small-scale mass balance (Golston et al., 2018; Gålfalk et al., 2021; Zondlo, 2021; Reuter et al., 2021), although not many sensors meet the precision as well as size, weight, and power requirements to deploy in this fashion. Long open-path measurements to a UAV were very recently used in conjunction with a Gaussian plume model to determine an emission rate from a point source (Soskind et al., 2023). Finally, several mass-balance approaches have been demonstrated using column-integrated measurements including solar-occultation flux (which can only be used during daytime/sunny conditions) (Mellqvist et al., 2010; Kille et al., 2017) and airborne LiDAR (which has focused on methane or carbon dioxide) (Ravikumar et al., 2019; Amediek et al., 2017; Bell et al., 2022; Kunkel et al., 2023; Johnson et al., 2021). There are two significant distinctions between these LiDAR approaches and the approach discussed here. First, the LiDAR systems are mounted on a larger aircraft, which has added cost and complications but does not require a van and can more easily cover a large area. Second, the LiDAR targets a single species, which is well suited to finding methane leaks in an oil/gas field, for example, while the system here relies on broadband dual-comb spectroscopy that can detect multiple species. If used in conjunction with a mid-infrared dual-comb system, this approach could then simultaneously detect multiple volatile organic compounds beyond methane.

Here, we demonstrate a new, mobile, micrometeorological mass-balance method using a line-integrated sensor (in this case, open-path dual-comb spectroscopy or DCS) in combination with an unmanned aerial vehicle. This is accomplished by measuring slant columns to a moving UAV that carries a small retroreflector downwind of an emission source as shown in Fig. 1. We test this technique using controlled releases of $CH_4$ and $C_2H_2$ from both a point source and a small area. A key strength of this technique – and other mass-balance techniques – is that it does not rely on a dispersion model. In principle this approach is also compatible with any open-path laser measurement, such as tunable diode laser spectroscopy (Plant et al., 2015; Bailey et al., 2017; Dobler et al., 2017; Bai et al., 2022), but here we use a frequency comb to allow for simultaneous multispecies detection (Coddington et al., 2016; Cossel et al., 2021; Picqué and Hänsch, 2019). For example, while $CH_4$ and

$C_2H_2$ measurements are demonstrated here, open-path dual-comb spectroscopy has been used to retrieve a host of interesting

species including $CO_2$, $NH_3$, HDO, ethane, formaldehyde (HCHO), CO, and $N_2O$ (Waxman et al., 2017; Ycas et al., 2019; Giorgetta et al., 2021; Herman et al., 2021).

## 2 Experimental setup

This method builds on our previous demonstration of spatially-scanned DCS (Cossel et al., 2017), which is summarized in Figure 1(a). The primary components of the system (Figure 1b) are the dual frequency combs, a transmit/receive terminal that

sends the light over a long open-air path, and a mobile reflector on a quadcopter.

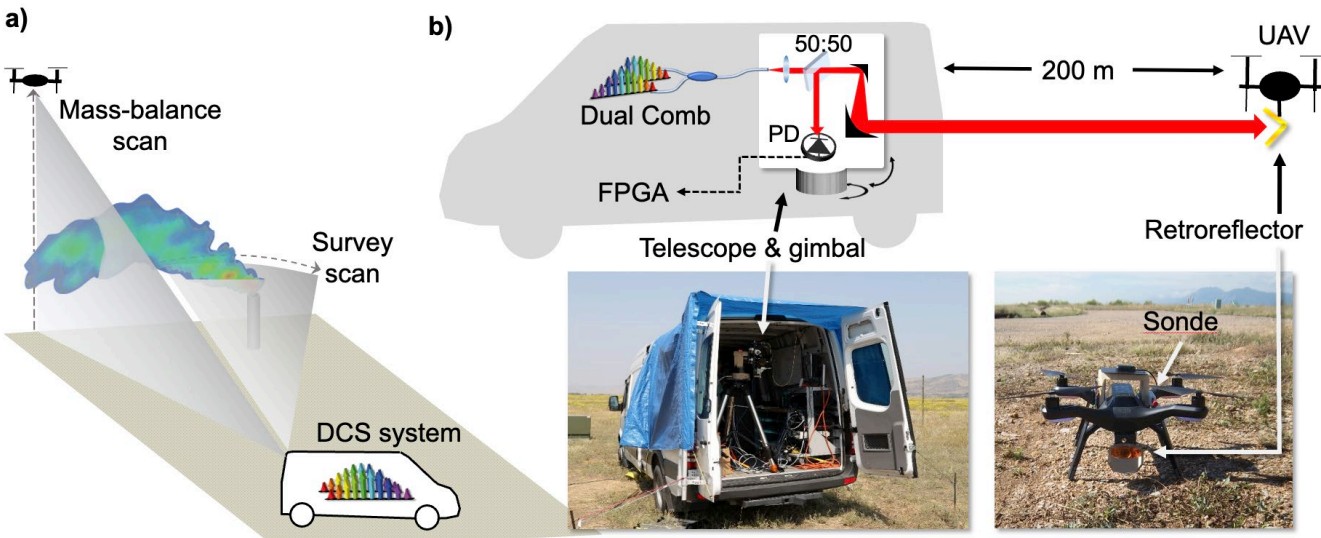

**Figure 1. (a) Overview of experimental concept. The DCS system measures integrated slant columns to a moving UAV. By performing survey scans, the presence and angular position of an emissions source can be rapidly determined. Mass-balance scans downwind of the source then enable quantification of the emissions. (b) Details of experimental setup. The light from dual frequency**

**combs is combined in fiber then launched out of a transmit/receive telescope system located on an azimuth/elevation gimbal to a retroreflector located on a UAV at a typical distance of 200 m. The return light from the retroreflector is separated with a 50:50 beamsplitter and measured on a photodetector (PD). This signal is digitized and averaged with a field-programmable gate array (FPGA)-based acquisition system. The lower part of (b) shows the van in the field as well as a photo of the UAV showing the retroreflector and sonde. (See also Appendix.)**

Here, we use robust Er:fiber-based frequency combs operating in the near-infrared (Waxman et al., 2017). The design and operation of these combs has been described in detail previously (Truong et al., 2016). Briefly, the frequency combs have a nominal repetition rate of 200 MHz and a repetition rate difference of $\delta f_{rep}$= 625 Hz. The combs are both stabilized to a crystal oscillator and cw reference laser at 1560 nm to maintain mutual coherence between the combs. After amplification, spectral

broadening, and spectral filtering to cover bands for $C_2H_2$ (1520 nm – 1540 nm) and $CH_4$ (1610 nm – 1670 nm), the light from the two combs is combined in fiber and then is launched from a transmit/receive telescope (76.2-mm-diameter aperture) to a

retroreflector (62.5-mm diameter) located on a UAV (here, a quadcopter). Return light from the retroreflector is collected with the same telescope and reflected off a 50:50 beamsplitter to a photodetector. Alignment to the retroreflector is maintained by an image-processing-based pointing servo using an 850-nm LED and Si CMOS camera co-aligned with the main telescope (Cossel et al., 2017). The return power was typically 100-200 µW with 10-20% power fluctuations between each measured spectrum as well as larger "dropouts" and power drifts due to alignment issues such as sudden UAV movement or yaw of the UAV. These dropouts were usually <10% of a single flight. Figure 1(b) shows a more detailed view of the UAV. In addition to the retroreflector, the UAV also carries several sondes to measure temperature, pressure, humidity, and GPS location, as well as real-time kinematic (RTK) GPS for high-precision relative GPS location of the UAV. However, sufficient location precision was provided by the on-board GPS, so that is used for most of the data presented here. For field operations, the DCS system is housed in a van and connected via fiber to a telescope and fast azimuth-elevation gimbal located at the back of the van as pictured in Figure 1(b). Wind speed and direction were measured with a 3D sonic anemometer located away from the van at ~2 m above ground. Additional meteorological parameters (temperature, pressure, independent wind speed and direction) were recorded by a weather station with a 2D sonic anemometer located above the van roof. This second wind measurement was used for redundancy and to verify the 3D sonic measurements. Finally, we measured $CH_4$ concentrations with a commercial cavity-ringdown spectrometer (CRDS) that sampled air from above the roof of the van. This spectrometer was calibrated at the NOAA Global Monitoring Laboratory to provide WMO-traceable $CH_4$ measurements, which we use to determine the background $CH_4$ concentration during flights.

The dual-comb signal on the receive photodetector is a time-domain interferogram (IGM) that repeats at a rate of $\delta f_{rep}$. We digitize this signal and co-add using a field-programmable gate array (FPGA). Sequential sets of 10,000 IGMs are coadded (with phase correction applied to each set of 100 sequential IGMs). In total this gives a 16s sampling period. In post processing, each saved IGM was converted to a transmission spectrum (Figure 2(a)) via a Fourier transform and then fit with a spectral transmission model (calculated from HITRAN2008 (Rothman et al., 2009) using the measured temperatures and pressures) plus a piecewise polynomial baseline term to determine total column densities for three gases, $CH_4$, $H_2O$, and $C_2H_2$ along the laser path, $\bar{\rho}_g = \int_0^L \rho(x,y,z)dr$, where $\rho_g(x,y,z)$ is the spatially varying density of the gas and $r$ is the position along the path (Waxman et al., 2017; Cossel et al., 2021). These total column densities are converted to path-averaged dry mixing ratios for $CH_4$ and $C_2H_2$ reported in ppm (µmol/mol) or ppb (nmol/mol) using the temperature and pressure from the weather station, the temporally varying path length determined from the UAV GPS location, and the DCS-measured water vapor for the dry-air correction (Waxman et al., 2017). The measurement precision is characterized by the Allan-Werle deviation of the retrieved path-averaged concentration during one flight with no gas release (thus approximately uniform mixing ratios) as shown in Figure 2(b). For this flight, the number of co-added interferograms was reduced to 2,500, resulting in measurements every 4 s. At the typical measurement time of 16 s, the $CH_4$ precision is around 23 ppm-m (57 ppb for a round-trip path length of 400

m). This performance is similar to that obtained from (Waxman et al., 2017) – 50 ppb extrapolated to the same measurement
time and path length – indicating very little degradation due to motion of the UAV.

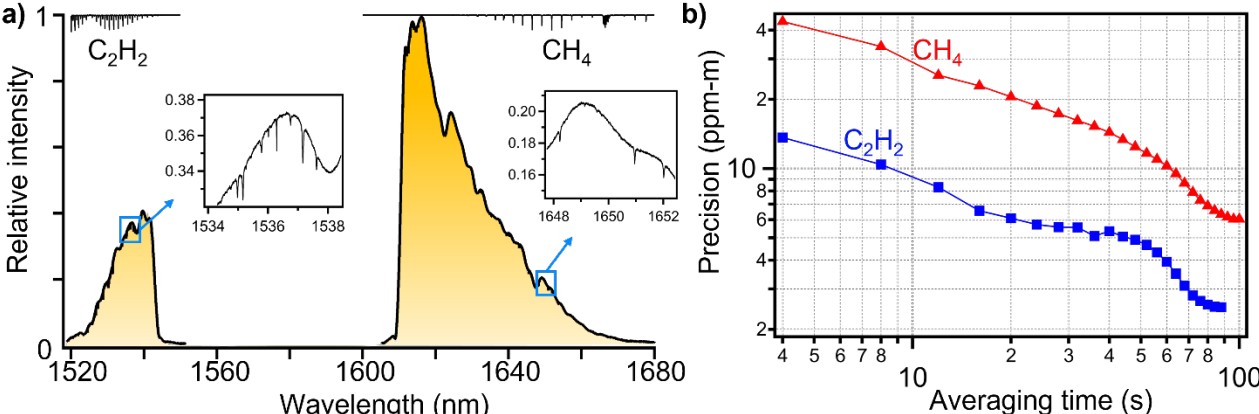

**Figure 2. (a) Example normalized transmission spectrum (10-minute average). C₂H₂ is retrieved in the band around 1540 nm and CH₄ is retrieved in the band around 1640 nm. A simulated transmission spectrum for both gases is shown along the top to guide the eye. The slower undulating structure in the spectra is stable over short times and results from the supercontinuum generation and**
**spectral etalons in the system. (b) Allan deviation of $\bar{\rho}_g$ for both CH₄ and C₂H₂ at ambient concentration from one flight. The round-trip path length was around 400 m.**

Flights were performed across four different days. Each flight had a maximum duration of ~10 minutes. Several different flight patterns were used; here we focus on horizontal and vertical scans. The UAV was manually piloted for all the flights. Several
different leak locations and arrangements were used in order to mimic emission sources that might be observed in the field. Controlled releases of CH₄ and C₂H₂ were located ~100 m from the van and were conducted using cylinders whose flow rate was either controlled with a mass flow controller or monitored with a flow meter (see Appendix for details). For CH₄, the flow rate was set around 0.22 g/s. For C₂H₂, the flow rate was set at around 0.18 g/s but was only used during a subset of releases. These flow rates correspond to a leak of ~0.7 kg/hr for methane, corresponding to a practical lower bound for systems detecting
methane leaks in oil/gas fields (Ravikumar et al., 2019; Johnson et al., 2021; Bell et al., 2022). The flow rate for acetylene was chosen to provide a similar signal level; future tests with multiple species would be coupled with the required species-dependent sensitivity. After the flow meter/controller, the gas was sent through a few-meter-long piece of PTFE tubing whose end was located between 0 m and ~5 m above ground level to simulate a point source emission. Small-area diffuse emissions were also simulated by puncturing the PTFE tubing every ~30 cm and placing the tubing on the ground across a ~2 m diameter
area.

## 3 Results

### 3.1 Survey scan for emissions detection

We first demonstrate rapid detection of an emission source and later quantification of its emissions. For detection, the UAV is scanned horizontally at 200-m distance from the telescope – as illustrated by the dotted grey line in Fig. 1(a) – resulting in measurements across a series of near-horizontal slant paths. For this demonstration, the source emitted both $CH_4$ and $C_2H_2$ from a pole ~5 m above ground level (AGL) located ~100 m away from the launch/receive telescope.

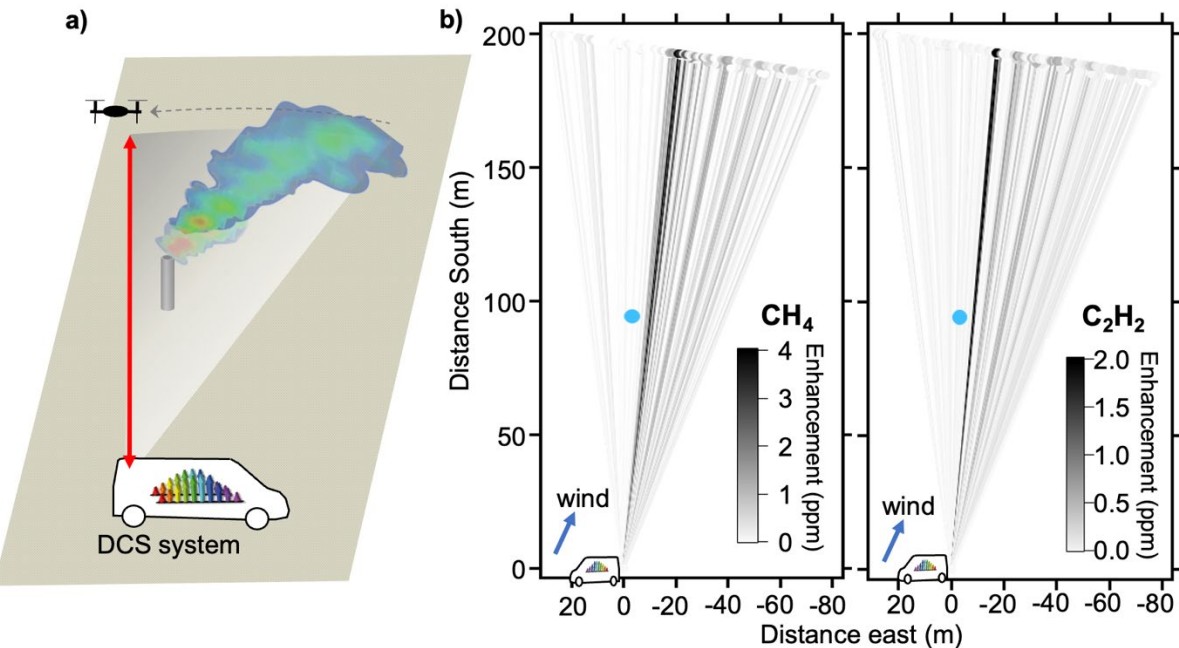

**Figure 3. (a) Detection and location concept using a constant altitude horizontal scan. (b) Map of the results for $CH_4$ (left) and $C_2H_2$ (right) from a flight with both $CH_4$ and $C_2H_2$ releases. The van is located at (0,0) and the UAV relative position is shown by the dots. For this flight, two horizontal scans were completed at 5 m and 9 m AGL. The release location is shown by a blue dot, and the wind direction shown by a blue arrow. The release was at a height of ~2 m AGL, and the mean wind speed was 2 m/s. Each measurement path is shown by a solid line, shaded by the path-averaged $CH_4$ or $C_2H_2$ enhancement over background (see text for details). For both gases, a sudden increase in the enhancement is visible ~5 meters downwind of the leak source.**

To analyze the results, we calculate the path-averaged enhanced column density ($\Delta\bar{\rho}_g$) for each gas $g$ ($CH_4$ and $C_2H_2$),

$\Delta\bar{\rho}_g = \bar{\rho}_g - L\langle\rho_g\rangle$, where $\bar{\rho}_g$ is the measured column density, $L$ is the path length, and $\langle\rho_g\rangle$ is the mean background concentration. For $CH_4$, the background concentration is obtained from the dry mixing ratio from the CRDS using the known air temperature, pressure, and water vapor interpolated to the DCS data timestamp. For $C_2H_2$, the background concentration is set to the mean of the DCS measurements without any release. Figure 3(b) shows the $CH_4$ and $C_2H_2$ path-averaged concentration enhancement ($\Delta\bar{\rho}_g/L$) from a single flight consisting of two horizontal scans at different altitudes (scans at both altitudes are aggregated in one plot). As expected, we observe significant enhancements of $CH_4$ and $C_2H_2$ for slant paths

when the UAV is immediately downwind of the emission location. The enhancements persist but decrease in amplitude as the paths move further downwind. Once the presence of an emissions source is detected with a horizontal scan, additional flights can be used for emissions quantification as discussed below. Further localization can be accomplished with more complex flight patterns (Soskind et al., 2023).

## 3.2 Mass-balance scan for emissions quantification

### 3.2.0 Methods

In order to perform emissions rate quantification, the UAV flew a vertical profile downwind of the emissions source as illustrated in in Figure 4(a). An example altitude profile recorded by the GPS on-board the UAV is shown as an inset. In this case, two vertical profiles between ~2 m and 30 m above ground level were performed during a single flight. Flights were also conducted with a single vertical scan during the flight. The flight patterns result in a series of measurements of the column density (or path-integrated concentration), $\bar{\rho}_g(z)$, for each gas species $g$ and UAV height $z$ with 16-s integration times.

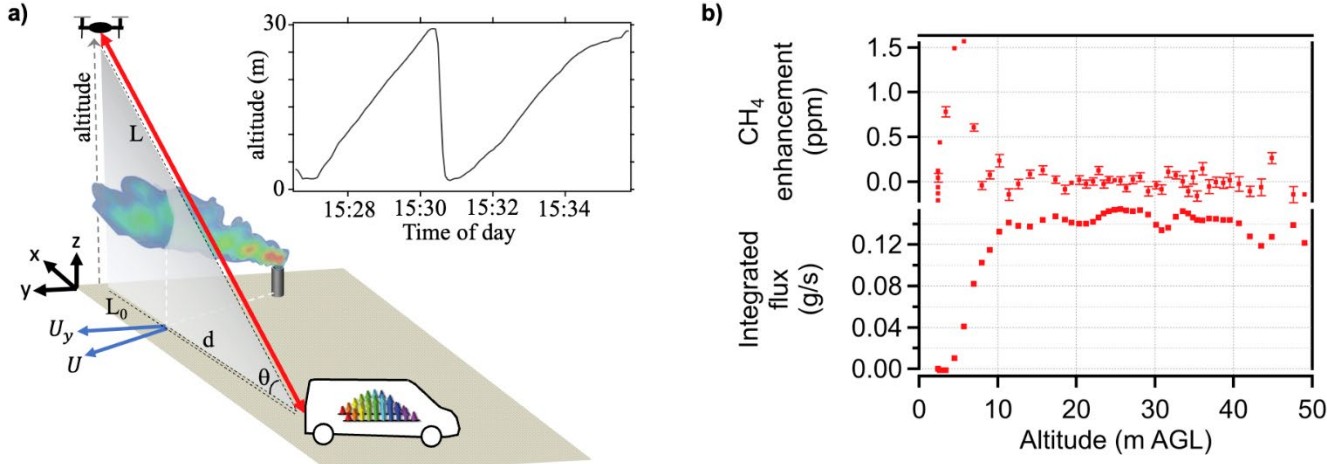

**Figure 4. (a) Diagram of measurement configuration for flux determination from and emission source. The UAV is located a distance $L$ from the DCS system and is scanned vertically with altitude above ground level given by $z$. The DCS system measures a slant column from the comb system to the UAV and back, as the UAV slowly changes altitude. The plume is transported by the wind vector denoted by $U$ (with the component perpendicular to the measurement plane given by $U_y$) and intersects the measurement plane a distance $d$ from the comb system. The inset shows an example UAV altitude profile (above ground level) for one flight. (b) Example data from one flight. (top panel) $CH_4$ enhancement above background versus UAV altitude. (bottom panel) Integrated flux versus altitude.**

To determine the flux, $F$, we start by following other mass-balance approaches (Alfieri et al., 2010; Karion et al., 2013; Mellqvist et al., 2010). With the geometry of Figure 4, we assume the x-axis is defined by the path from the telescope to the UAV position projected on the ground, and the z-axis is vertical. We can then write the flux through a closed surface $S$ as

$$F = \oiint_S \Delta\rho(x, y, z)\boldsymbol{U} \cdot d\boldsymbol{A}, \tag{1}$$

where $\boldsymbol{U}$ is the wind velocity vector, with incremental area $d\boldsymbol{A}$ is the incremental area of the surface, and $\Delta\rho(x, y, z)$ is the gas density above background. If we assume a significant y-component to the wind velocity vector, $U_y$, then the entire plume passes through the x-z plane. Thus, the total flux is found by integrating the enhanced concentration across the x-z plane or

$$F = \iint \Delta\rho(x, z)U_y dx dz. \tag{2}$$

However, because the DCS system is measuring a slant column between a fixed point and a moving point – $i.e.$, it measures $\Delta\bar{\rho}(r, \theta) = \int_0^L \Delta\rho(r, \theta)dr$ – we convert Equation (2) to polar coordinates and restrict the integrals to the plane shown in Figure 4(a) ($0 \le \theta \le \theta_{max}$), assuming that the end position of the UAV in both x and z coordinates lies beyond the plume. Then,

$$F = \int_0^{\theta_{max}} U_y \int_0^L \rho(r, \theta)r dr d\theta \tag{3}$$

Note the presence of the additional $r$ term in the integral, which means that this integral is not directly what the DCS system measures. To evaluate this, we assume that $\rho(r, \theta) = \Delta\bar{\rho}(\theta)\delta(r - d/\cos\theta)$. That is, we assume the plume is localized to intersect the measurement plane at $x = d$. Then,

$$F = \int_0^{\theta_{max}} U_y \int_0^L \Delta\bar{\rho}(\theta)\delta\left(r - \frac{d}{\cos\theta}\right)r dr d\theta. \tag{4}$$

Evaluating the radial integral gives

$$F = \int_0^{\theta_{max}} U_y \Delta\bar{\rho}(\theta)\frac{d}{\cos\theta} d\theta. \tag{5}$$

Converting back to cartesian coordinates with $z = L_0 \tan\theta$ and assuming that $z \lesssim L/4$ gives finally

$$F \approx \frac{d}{L_0}\int_0^H U_y \Delta\bar{\rho}(z)dz, \tag{6}$$

where $H$ is the maximum altitude. Equation (6) is derived for the delta-function plume but is valid for other plume shapes where $d$ is defined using the mass-weighted mean. In addition, it assumes that $d$ and $L_0$ are constant through one flight. We note the $d/L_0$ correction is unnecessary if both the launch and reflector could be moved together, in which case Equation (2) can be evaluated directly. We can intuitively understand the presence of the $d/L_0$ by looking again at a narrow plume intersecting the measurement plane at $x = d$. The effective altitude range at the plume is $\frac{d}{L_0}H$ instead of $H$, thus we need to do a change of variable to $z' = \frac{d}{L_0}z$ when doing the altitude integral.

To calculate the flux from a vertical scan, we first interpolate the auxiliary data (UAV location, wind speed and direction, CRDS CH$_4$) to the DCS data timestamp and filter the DCS data for low signal-to-noise ratio (for example, if the telescope tracking lost alignment briefly). Then, we determine $\Delta\bar{\rho}_g$ as in Section 3.1. Finally, Equation (6) is numerically integrated to obtain the total flux.

The result of the numerical integration of Equation (6) versus altitude is shown in the top panel of Fig. 4(b) for one flight. For this flight, the $CH_4$ enhancement profile shows a clear peak between 0 m and 10 m AGL, which corresponds to a rapid increase in the integrated flux. After 10 m AGL, the enhancement has dropped back to background levels and remains near background up to 50 m AGL. As expected from this enhancement profile, the integrated flux shows a steady increase from 0 m to 10 m AGL, after which it remains relatively constant up to 50 m. The variations between 10 m and 50 m are driven by measurement noise and provide an estimate of the sensitivity of the flux determination. Note that any offset between the CRDS background and the DCS background would lead to a linear slope during this period from 10 m to 50 m, which is not observed in the measurements. From these data, it is also clear a future system could dispense with the separate CRDS sensor at the van and instead use the flat high-AGL measurements or measurements taken upwind of the source by, e.g., combining the scans in Figure 3 and 4.

### 3.2.1 Results

Flights were performed across four different days (see Appendix for details of the flights used). Approximately 20 vertical profile flights were conducted; however, on several flights, the wind direction was wrong or shifted early in the flight and caused the plume to miss the measurement path. We note that this information is known in the field from of the meteorological sensors, and the measurement can simply be repeated when the wind has stabilized. In addition, on two flights, the integrated flux was still increasing at the final altitude, indicating that slant path did not reach the top of the plume, so these flights were discarded from the analysis. In total, 16 flights had sufficient data to determine a flux. Of these, four flights were performed with no release and were evaluated to determine "background" flux. Figure 5 shows a summary of the $CH_4$ flux determined for these flights.

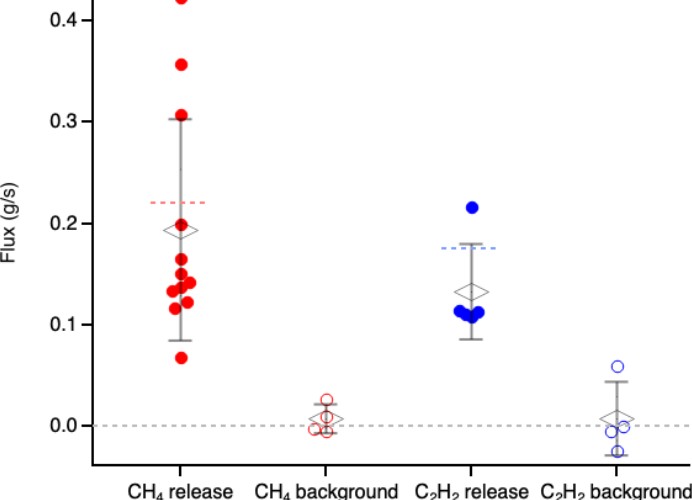

**Figure 5. CH₄ and C₂H₂ flux determination for flights with a release and without a release ("background"). A single flight corresponds to a single point for CH₄ and for C₂H₂. For each set, the mean is shown by a diamond and the standard deviation indicated by the vertical lines. The CH₄ release rate was 0.22 g/s and the C₂H₂ rate was 0.18 g/s as indicated by the dashed horizontal lines.**

For the 12 flights with a CH₄ release, the mean flux determined was $0.19 \pm 0.11$ g/s to within one standard deviation. The mean agrees within 15% of the expected flux of 0.22 g/s, while the standard deviation indicates 56% uncertainty for a single flight. Of these 12 flights, four flights had a point release and eight flights had a diffuse release, illustrating the capability to measure both types of emissions. No obvious differences were observed for the two release types. For these flights, the distance of the measurement plane downwind from the source varied from ≈10 m to ≈60 m. The four background flights give a mean of 0.008±0.014 g/s. We estimate the detection limit for CH₄ fluxes as twice the deviation of these background flights, or about 0.03 g/s. For five flights, C₂H₂ was also released at 0.18 g/s with measured flux values of $0.15 \pm 0.05$ g/s, again in good agreement with the release rate. The C₂H₂ background flights give a mean of 0.008±0.04 g/s.

### 3.2.2 Discussion

The uncertainty in the flux determination is a result of several contributions. First, we note the accuracy of the methane gas concentration measured by DCS is at the level of ~ 1% (Waxman et al., 2017) and is not a significant contribution. The uncertainty arises in the conversion of these measurements to a flux determination via Equation (6) due to several effects. First, incorrect measurement of the background gas concentrations (for example, due to biases between two CH₄ measurement systems) could lead to an error in $\Delta\bar{\rho}_g$ and a corresponding error in the flux. However, such a bias will result in a linear slope even without a gas flux. From the data in Figure 4(b) for altitudes >10 m, we see no evidence of a linear slope. This is also true in the background flights. Again, this is not surprising given the accuracy of both DCS and the cavity ringdown spectrometer used for the background gas measurements, and we conclude the evaluation of $\Delta\bar{\rho}_g$ is not a significant source of uncertainty.

Errors in $U_y$ will lead directly to errors in the determined flux. We estimate an upper limit on this error by comparing the two different wind sensors. Their values of $U_y$ averaged over a 10-minute flight agree to within ±20% and typically better. The difference is dominated by uncertainty in the wind speed, rather than the wind direction, given the geometry chosen here where the plume is approximately normal to the measurement plane. (The uncertainty in the wind direction is however important below in the determination of $d$.) The difference in $U_y$ between the anemometers is likely due to true wind differences at the location of the wind sensors. It is possible that a UAV-based wind measurement could reduce this error and improve the flux determination.

There are three sources of uncertainty associated with the value of $d$ in Eq (1). First, errors in the source location will cause errors in the determination of $d$. Assuming a random source location uncertainty of $\Delta x$, the corresponding uncertainty in $d$

will also be $\Delta x$. For the measurements here, the source location was known to better than 1 m, so this uncertainty is negligible. In the case of an initially unknown source location, the approximate location will need to be determined well enough to meet the target accuracy goals based on the measurement configuration. For example, to keep this uncertainty below ±20%, the source location needs to be known to within ±$0.2d$. For the flights here with $d \approx 100$ m, the location would need to be known within ±20 m. In the case of a uniform diffuse source, the effective weighted center of the source needs to be known to this level. Second, even with a known source location, uncertainty in the wind direction $\Delta\phi$ will also lead to an uncertainty in $d$. There are two potential sources of wind direction errors: a static bias during one measurement between the measured wind direction at the sensor and the actual wind direction along the plume trajectory, and a temporally varying difference between them. A static bias over one flight will lead to an error given by $\Delta\phi \times d_{dw}$, where $d_{dw}$ is the downwind distance between the source and the measurement plane. Following the geometry here, the downwind distances are <50 m, so that a ±30° wind direction error (estimated from the mean difference between the two anemometers during different flights) corresponds to ±25 m error in $d$ or ±25% error in the flux for $d = 100$ m. To investigate the impact of the within-flight variability, we recalculated the flux using a time-varying value of $d$ (i.e. mapping $d$ to a slowly varying function $d(z)$ within Eq. (6) using the known values of UAV altitude $z$ as a function of time). For the 10 $CH_4$ release flights, the change in flux was in all cases <13%, with a change in the mean value of 1.4%, indicating that the impact of temporally varying wind direction differences is minimal. As mentioned above, the static bias could be reduced by measuring the wind direction at the UAV itself. Furthermore, these wind direction related errors are minimized with longer measurement paths, which also help to reduce the impact of source location uncertainties. Third and finally, the use of $d$ in Eq (1) relied on the assumption the weighted centroid of the plume followed the wind direction from the source to the measurement plane. Plume dynamics are complex and there is some inherent uncertainty in the plume evolution over time. As an estimate of this effect, we assume that the plume centroid is offset on average by at most $\sigma_y$ from the expected location based on the mean wind direction. The value of $\sigma_y$ can be taken as the average beam spread in a Gaussian plume model (Seinfeld and Pandis, 2006). For typical daytime conditions with high solar insolation and ~2 m/s wind speeds (stability class 'A'), $\sigma_y \approx 15$ m, which corresponds to a ±15% error for $d = 100$ m. The combination of the assumed 20% uncertainty from the source location, 25% uncertainty from the wind direction, and 15% uncertainty from the plume location yields a total ~35% uncertainty related to $d$.

Finally, Equation (2) implicitly assumes the plume location is fixed over the measurement. However, if the vertical position of the plume changes during the measurement, then we have not truly measured the instantaneous flux of Equation (2). Assuming a flight with vertical velocity, $V$, and a time-dependent concentration $\Delta\rho(x, z, t)$, the actual measured quantity is

$$F' = \iint \Delta\rho(x, z, V^{-1}z)U_y dx dz, \tag{7}$$

where we ignore the corrections due to the slanted path since they have already been discussed. To estimate the resulting error from vertical translation of the cloud during the measurement, we use a gaussian plume model,

$$\Delta\rho(x, z, t) = \frac{U_y F}{2\pi\sigma_x\sigma_z} \exp\left(-\frac{(x-x_0)^2}{2\sigma_x^2}\right) \exp\left(-\frac{(z-z_0(t))^2}{2\sigma_z^2}\right), \tag{8}$$

where the centroid of the two-dimensional plume position is given by $(x_0, z_0)$ with corresponding widths $\sigma_x$ and $\sigma_y$. We write the slowly varying time dependence of the vertical position as $z_0(t) = z_0 + \delta z_0(t)$. To lowest order, the vertical position changes due to a small average vertical wind velocity component, $U_z$, over the roughly two-minute measurement time, giving $\delta z_0(t) = U_z t$ where $t = V^{-1}z$. Substitution of (8) into (7) and a Taylor expansion about $\delta z_0$ yields the fractional error of the measured quantity compared to the desired flux,

$$\left|\frac{F'}{F} - 1\right| = \frac{U_z}{\sqrt{2\pi}\sigma_z^3 V} \int z(z - z_0) \, exp\left(-\frac{(z-z_0)^2}{2\sigma_z^2}\right) dz = \frac{U_z}{V} \tag{9}$$

For our flights, we had chosen a relatively slow vertical velocity of $V \approx 0.2$ m/s. Based on measurements from the 3D anemometer, we find a typical vertical wind speed of ~0.05 m/s giving an error of ~25%.

As shown by the background flights, the uncertainty due to DCS measurement noise is expected to contribute ±6% for a $CH_4$ flux of 0.22 g/s. So, when combined, the ±20% uncertainties associated with the values of $U_y$, ±35% uncertainty in $d$, and the ±25% uncertainty from a time-dependent $z_0$, we estimate a total estimated uncertainty of ~±50% in the measured flux values for a single flight, which is in good agreement with the observed uncertainty. Finally, we note that the uncertainties associated with $U_y$, $d$, and $z_0$ described above are statistical (driven by atmospheric variability and plume dynamics), thus are expected to average down with multiple measurements, as observed here with the low mean bias in the flux determination.

## 4 Conclusions

We have demonstrated a rapid mass-balance method for flux determination using path-integrated slant column measurements between a ground-based measurement system and a UAV located at varying heights above ground. Using a near-infrared dual-comb spectroscopy system, we show flux quantification of $CH_4$ to within 50% with a single <10 min flight and an estimated detection limit of 0.03 g/s (2 sigma), which would enable detection of emissions from 25 head of cattle or from a single pneumatic controller. It is estimated that 90% of all oil field emissions come from sources that are ten times larger than this limit (Brandt et al., 2016). We can also simultaneously determine $C_2H_2$ fluxes with similar performance. This study was designed as a proof of concept for the method. Due to available resources and other logistical considerations, as well as some equipment malfunctions, the measurements were limited in scope. The next step is to do more extensive testing over a range of release conditions, for example, at a facility such as the Methane Emissions Technology Evaluation Center (METEC) (Edie et al., 2020; Riddick et al., 2022).

This new methodology has several potential advantages compared to other flux measurement methods. First, a key advantage is that no atmospheric dispersion model is needed since the flux is determined directly from the data. This also means that multiple spatially separated sources or areal sources can also be measured, although in an area with many sources, care needs to be taken so that the background is appropriate for the sources of interest. A limitation of the specific flight pattern shown is that the source location needs to be approximately known; however, this limitation can be overcome either with *a priori*

information (e.g., if the equipment or facility to be measured is known) or by performing spatial scans first. In addition, modified flight patterns such as flying vertically and then horizontally toward the source could likely overcome this limitation. Second, the methodology is flexible, so it can be used to determine fluxes for any gas that can be measured with open-path dual-comb spectroscopy or other open-path spectroscopy such as active differential optical absorption spectroscopy (Stutz et al., 2016). In particular, $CO_2$, $NH_3$, HDO, ethane, formaldehyde (HCHO), CO, and $N_2O$ have all been measured with DCS

(Waxman et al., 2017; Ycas et al., 2019; Giorgetta et al., 2021; Herman et al., 2021). By only requiring the lightweight retroreflector to be flown, a small UAV can be used regardless of the gas or gases to be measured.

Combined, these advantages give the capability for rapid, easily deployable, multispecies flux measurements from point or distributed sources. This could be beneficial for example to survey emissions from fields, agricultural facilities, wastewater

treatment plants, and oil and gas facilities. In addition, with further engineering of mobile DCS, measurements could be conducted from a moving van (similar to the solar occultation flux technique (Mellqvist et al., 2010)), allowing for flexible and rapid coverage of a wide area.

*Data Availability.*

Data available from the authors on request.

*Author Contributions.*

K.C. and E.W. collected and analyzed the data. E.H. assisted with the field measurements and data collection from the UAV. D.H. and C.C. piloted the UAV. K.C., E.W., I.C., and N.N. wrote and edited the manuscript.

*Competing Interests.*

The contact author has declared that none of the authors has any competing interests.

*Acknowledgements.*

We thank Michael Cermak for technical assistance, Tim Newberger and Kathryn McCain from NOAA for assistance with the point sensor calibration, and Nathan Malarich, Ryan Cole, and Jerome Genest for helpful discussions on the manuscript.

*Financial Support.*

National Institute of Standards and Technology (NIST); DARPA Defence Sciences Office.

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

**5 Appendix: Details on the release conditions**

Flights were conducted on seven different days at the Table Mountain facility near Boulder, CO (40.13112, -105.24065). However, only flights from four days were able to be used here due to technical issues. One each day, the specific release location was chosen based on the local wind and was sometimes moved if the wind shifted. Flights occurred during daylight, from approximately 1000 – 1700 local time (Mountain Daylight Time). Between 5 and 10 flights were flown per day. Several 485 different flight patterns were flown including the vertical scans used here as well as some flights based on a vertical-radial plume mapping approach; however, as those flights only contained a few vertical steps, they could not be used for the flux analysis here. Table 1 lists the flights that were used as well as a summary of meteorological conditions for each flight.

| Flight # | Day | Leak type | Gas | Wind direction (deg) | Wind direction std (deg) | Wind Speed (m/s) | Wind Speed std (m/s) | Temp (C) | Solar |
|---|---|---|---|---|---|---|---|---|---|
| 1 | 10/4/2017 | Point | CH4 | 40 | 20 | 1.6 | 0.5 | 17.85 | sunny |
| 2 | 10/4/2017 | Point | CH4 | 57 | 37 | 1.4 | 0.6 | 17.85 | sunny |
| 3 | 10/4/2017 | Point | CH4 | 54 | 31 | 1.4 | 0.6 | 18.05 | sunny |
| 4 | 10/4/2017 | Point | CH4 | 36 | 20 | 2.3 | 0.5 | 18.15 | sunny |
| 5 | 10/13/2017 | Distributed | CH4 | 101 | 28 | 1.4 | 0.5 | 15.55 | partly cloudy |
| 6 | 10/18/2017 | Distributed | C2H2 & CH4 | 127 | 23 | 2.3 | 0.7 | 24.45 | sunny |
| 7 | 10/18/2017 | Distributed | C2H2 & CH4 | 103 | 14 | 2.8 | 0.6 | 24.35 | sunny |
| 8 | 10/18/2017 | Distributed | C2H2 & CH4 | 129 | 9 | 2.5 | 0.5 | 24.15 | sunny |
| 9 | 10/18/2017 | None | None | 124 | 12 | 2 | 0.5 | 23.95 | sunny |
| 10 | 11/3/2017 | None | None | 123 | 51 | 1.1 | 0.4 | 5.55 | low clouds |
| 11 | 11/3/2017 | Distributed | C2H2 & CH4 | 78 | 29 | 1.8 | 0.7 | 5.75 | broken low clouds |
| 12 | 11/3/2017 | Distributed | C2H2 & CH4 | 78 | 21 | 2.1 | 0.6 | 6.05 | partly cloudy |
| 13 | 11/3/2017 | None | None | 117 | 34 | 2.3 | 0.9 | 7.35 | partly cloudy |
| 14 | 11/3/2017 | Distributed | CH4 | 113 | 29 | 2.1 | 0.7 | 7.95 | partly cloudy |
| 15 | 11/3/2017 | Distributed | CH4 | 80 | 44 | 1.2 | 0.6 | 8.45 | partly cloudy |
| 16 | 11/3/2017 | None | None | 144 | 35 | 1.8 | 0.6 | 9.95 | partly cloudy |

Table 1: Release information for the processed flights.

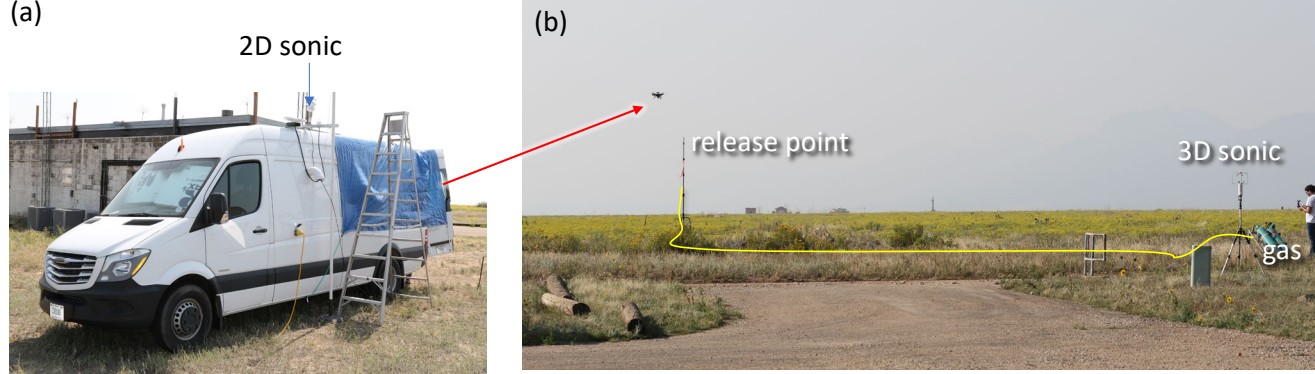

Figure 6: Photo of the setup for the controlled releases. (a) The van that housed the dual-comb spectrometer with a 2D sonic anemometer/weather station (Gill GMX500) mounted to its roof. (b) Standard gas cylinders provided methane and/or acetylene at >99% purity. The flow was set by an Alicat MC-20SLPM flow controller with a specified accuracy of +/- 0.6%, 495 after which PTFE tubing led to the release point, as discussed in the text. Gas flow for the second gas was set with a ball valve and monitored with a float flow meter (estimated accuracy +/- 10%) A second 3D anemometer (RM Young Model 81000 3-axis ultrasonic) provided three-dimensional wind data. The uncertainties in the flow rates were negligible compared to the larger total 50% uncertainties discussed in the text. As can be seen, the terrain was quite flat with a low cover of grass and brush (<1 m high) typical of the western United States.
