# Peer review of "Ground-to-UAV, laser-based, emissions quantification of methane and acetylene at long standoff distances"

_EGUsphere, 2023_

## Author Comment (AC1)

**Response to Reviewers regarding "Ground-to-UAV, laser-based, multi-species emissions quantifications at long standoff distances" by Cossel et al., egusphere-2023-691**

**Response to Anonymous Referee #1:**

The work by Cossel at al. "Ground-to-UAV, laser-based, multi-species emissions quantifications at long standoff distances" is well written, is a relevant contribution, and the topic discussed is within the scope of AMT. I recommend publication after minor revisions. However, the title "Ground-to-UAV, laser-based, quantifications of CH4 and C2H2 at long standoff distances" would more adequately describe what is presented (as the authors correctly state, the method is certainly expandable to a variety of species, nevertheless this is an extrapolation, while the title should as accurately as possible depict what actually is delivered).

We thank the referee for their helpful comments and have addressed their suggested revisions below. We have adjusted the title to one very similar to that suggested: "Ground-to-UAV, laser-based, emissions quantification of methane and acetylene at long standoff distances".

Comments:

Not much detail is provided on the quality of the recorded interferograms and of spectra derived from these. It would be interesting to give the reader a feeling of the level of degradation introduced by applying DCS in the open field with UAV-borne retroreflector. Specifically, I would be interested to learn how variable the interferograms are due to variable coupling efficiency. The spectral envelope appears surprisingly structured (fig 1a): are these undulations and the overall spectral intensity level variable from spectrum to spectrum?

[KCC]

The strength of the interferograms varies with the return signal strength from the UAV, which in turn depends on how well the telescope follows the UAV (so that the outgoing beam is fully incident up on the retroreflector) and whether the retroreflector orientation remains within the ~30-degree acceptance angle with respect to a line of sight to the van (beyond this angle, the return power begins to drop). Typical power fluctuations between each coadded interferogram were around +/- 10-20% for a 16-s coadd period with larger "dropouts" and power drifts due to alignment issues. With faster codding (1.6-4s), the interferogram-to-interferogram power fluctuations were closer to +/- 50%. We have one data set where we switched between a static retroreflector to the UAV. In this case, the average power was similar between the two cases, but the power fluctuations were ~5x larger (at 1.6 s averaging time) when using the UAV. To remove the dropouts, coadded data corresponding to very low power levels were not included in the analysis. These dropouts were typically < 10% of the total data. We have added the following comments to the Section 2:

Alignment to the retroreflector is maintained by an image-processing-based pointing servo using an 850-nm LED and Si CMOS camera co-aligned with the main telescope (Cossel et al., 2017). *The return power was typically 100-200 mW with 10-20% power fluctuations between each measured spectrum as well as larger "dropouts" and power drifts due to alignment issues such as sudden UAV movement or yaw of the UAV. These dropouts were usually <10% of a single flight.* Figure 1(b) shows a more detailed view of the UAV.

The undulating structure in the spectrum of Fig 1a is a result of structure in the supercontinuum generation as well as etalons in the optical system and is constant from spectrum to spectrum even with large power fluctuations. We have added the following sentence to the caption of Figure 1:

*The slower undulating structure in the spectra is stable over short times and results from the supercontinuum generation and spectral etalons in the system*.

The background concentrations are established in different ways for CH4 and C2H2. What is the advantage of introducing an additional separate sensor for measuring the background? (1) The use of two different sensors will always introduce some level of bias (I understand that the CRDS was calibrated properly, but I would expect a calibration bias of the CH4 band intensity reported in HITRAN 2008 in the range of 1 … 2%). (2) The UAV needs to climb up beyond the plume signal for a useful flux measurement anyway, and in a complex terrain covered with different sources, I would expect the CH4 concentration measured at a higher altitude to provide a more reliable background value than a measurement taken near ground.

Yes, we agree that the separate sensor is probably unnecessary. As the reviewer points out, we could as well have used the level at high AGL. One could also use a second, fixed open-path DCS provided that path did not intersect any plumes. Here we wanted the additional verification provided by the external sensor, and we wanted to be able to track background CH4 variation during a flight. (However, these variations were found to be minimal). We have addressed this in the new text at the end of Section 3.2.0:

*From these data, it is also clear a future system could dispense with the separate CRDS sensor at the van and instead use the flat high-AGL measurements as a background reference or measurements taken upwind of the source by, e.g., combining the scans in Figure 3 and 4.*

As shown in Figure 5, while the CH4 result is based on reasonably sound statistics (10 release results, 4 background results), there are only two C2H2 release results (and 4 background results). Why there are only 4 C2H2 background results? Could not all flights performed without C2H2 release (so 8 flights) be used for deriving C2H2 background values? Why have only two C2H2 release experiments been conducted?

We agree with the reviewer that the C2H2 release results are indeed limited. Unfortunately, this was due to several different factors. First, the goal of the measurement series was to show the technique for a single gas and C2H2 was added just to highlight the multispecies capability. Due to logistical and safety considerations, we had only a limited amount of C2H2 available. Several flights were not able to be used because the wind shifted rapidly during the flight. Some flights were also performed using a different flight pattern based on the vertical -radial plume mapping approach; however, these flights were not suitable for use in this analysis. Finally, we also had to discard a number of flights due to issues with a GPS data that were unfortunately only discovered in the post processing steps.

However, we did look back through the data and found couple of additional flights that had useable data. We now have a total of 12 CH4 release results and 5 release results. The corresponding data values and plots have been updated. The new figure is reproduced here:

[Figure]

Indeed, the reviewer is correct that all flights without the C2H2 release could be used to derive the C2H2 background. For consistency with the CH4 background analysis, we have chosen not to do that.

I am not sure to fully understand the discussion of plume dynamics (lines 270 - 275). Is this "plume centroid offset" an elongation along the horizontal or along the vertical? If I understand correctly, it is interpreted as a horizontal elongation resulting in an uncertainty in d. I would imagine that both horizontal and vertical variations during the measurement process are important, as the concentrations used for evaluating the integral along the altitude coordinate (equation 6) are actually not measured simultaneously, but during ~3 min (as I would estimate from fig 1a), so while the undulating plume is passing by. For further quantification of this source of uncertainty, performing a longer measurement with the UAV resting at an altitude corresponding to the average plume height would be informative.

Yes, the reviewer is correct that we had considered the horizontal elongation and the resulting uncertainty in *d*. We had not evaluated any additional uncertainty resulting from plume dynamics in the vertical direction that occur during the measurement itself. (For example, in a worst-case scenario that a narrow plume moved vertically upward as the UAV moved upward.) We thank the reviewer for pointing out this omission as the uncertainty is not negligible and could explain the previous discrepancy between our measured and estimated uncertainty. We have added the following uncertainty analysis to Section 3.2.2:

*Finally, Equation (2) implicitly assumes the plume location is fixed over the measurement. However, if the vertical position of the plume changes during the measurement, then we have not truly measured the instantaneous flux of Equation (2). Assuming a flight with vertical velocity, V, and a time-dependent concentration $\Delta\rho(x,z,t)$, the actual measured quantity is*

$$F' = \iint \Delta\rho(x,z,V^{-1}z)U_y dx dz, \tag{7}$$

*where we ignore the corrections due to the slanted path since they have already been discussed. To estimate the resulting error from vertical translation of the cloud during the measurement, we use a gaussian plume model,*

$$\Delta\rho(x,z,t) = \frac{U_yF}{2\pi\sigma_x\sigma_z} exp\left(-\frac{(x-x_0)^2}{2\sigma_x^2}\right) exp\left(-\frac{(z-z_0(t))^2}{2\sigma_z^2}\right), \quad (8)$$

*where the centroid of the two-dimensional plume position is given by $(x_0, z_0)$ with corresponding widths $\sigma_x$ and $\sigma_y$. We write the slowly varying time dependence of the vertical position as $z_0(t) = z_0 + \delta z_0(t)$. To lowest order, the vertical position changes due to a small average vertical wind velocity component, $U_z$, over the roughly two-minute measurement time, giving $\delta z_0(t) = U_z t$ where $t = V^{-1}z$. Substitution of (8) into (7) and a Taylor expansion about $\delta z_0$ yields the fractional error of the measured quantity compared to the desired flux,*

$$\left|\frac{F\prime}{F} - 1\right| = \frac{U_z}{\sqrt{2\pi}\sigma_z^3 V} \int z(z - z_0) exp\left(-\frac{(z-z_0)^2}{2\sigma_z^2}\right) dz = \frac{U_z}{V} \quad (9)$$

*For our flights, we had chosen a relatively slow vertical velocity of $V \approx 0.2 m/s$. Based on measurements from the 3D anemometer, we find a typical vertical wind speed of ~0.05 m/s giving an error of ~25%.*

*As shown by the background flights, the uncertainty due to DCS measurement noise is expected to contribute ±6% for a $CH_4$ flux of 0.22 g/s. So, when combined, the ±20% uncertainties associated with the values of $U_y$ , ±35% uncertainty in d, and the ±25% uncertainty from a time-dependent $z_0$, we estimate a total estimated uncertainty of ~±50% in the measured flux values for a single flight, which is in good agreement with the observed uncertainty.*

Minor technical comments / corrections:

In my feeling, it would be useful to provide (in a short appendix) some more detail on the "four different days" mentioned in section 3.2.1. Please provide the actual dates, some relevant details on the location (character of area, surface roughness), and some information on meteorological conditions during flights (average wind speed, variability of wind direction).

We have added an appendix with the requested information including a summary table and an additional photo of the site.

Line 241 typo "will lead to lead directly to"

Thank you. Fixed.

Line 277 " … a flux of 0.22 g/s." -> " … a CH4 flux of 0.22 g/s." (as measurement noise level is species dependent)

Thank you. Fixed.

**Response to Anonymous Referee #2:**

This is a well put together and concisely written article that demonstrates the use case for this type of methane and ethane emission measurement. It is a shame that the controlled release to demonstrate the use is so limited in scope as it would have been fantastic to understand the limitations in terms of limit of detection and how uncertainties trend with increased emission rates. However, as these experiments are now complete, I expect I will have to hope for an expanded experiment for this to be demonstrated (If there are more data from these experiments, please can this be included!).

In my opinion, this paper can be published with relatively minor corrections / clarifications. Most of the concerns regard full quantification of the uncertainty and lack of detail around steps in the experimental process.

We thank the referee for their comments. Yes, the usual experimental constraints limited the scope of the measurements, but we hope this manuscript will spur additional development. Due to logistical considerations as well as technical issues that we determine after the flights, the experiments here were limited in scope compared to the much larger range of test cases available at coordinated test sites, such as those used in the tests at METEC. As the reviewer suggests, we do plan to do a expand experiment in the future now that we were able to demonstrate to initial concept. We have addressed the referee's suggestions below.

1 The experimental set up for the controlled release.

L120 onwards.

There is scant description surrounding the controlled release. There needs to be sufficient detail in the methodology so that this experiment could be repeated. My recommendation would be to have a SI with the description of the controlled release in detail, potentially including photos, set-up, details of the equipment used, gas compositions, uncertainties inherent in the set up (and how they may propagate uncertainties through to the measurement results). Understanding how controlled releases are set up is an important component of being able to understand how any type of flux measurements are verified.

We have added this information to the appendix in the form of a Table and added figure that details the site conditions requested by Referee #1.

1 Characterization of the background

L146. From an operational perspective when using this system outside of a controlled release environment are there issues with assuming that the upwind pathlength is equal to the static measurement and therefore correlating the upwind line to that point measurement? I'm wondering if you could comment on the sensitivity of that pathlength to "unaccounted for methane" and how much methane could be present above baseline without the appearance of an apparent plume. This concerns me a little in settings where multiple plumes may be present and interfering with measurements or just unknown plumes from other sources.

Yes, as the reviewer point out, there are additional considerations when using this approach outside of a controlled release. Determination of the background is a key consideration, much like any mass balance approach. Any gas that intersects the measurement path but is not detected in the background will be interpreted as an enhancement and thus will be integrated as a flux. This can be either an advantage, since it allows complex sources to be measured, or a challenge if trying to isolate a single source among a mixture of sources. As discussed above in response to a similar question from reviewer 1, one could replace the point sensor measurements with open-path measurements to the UAV at high AGL or along a separate ground path, depending on the site location. In the future, one could also envision using two UAVs flying simultaneously to measure both a background and downwind plane. We discuss the issue of multiple plumes in response to the next question.

1 Influence of multiple plumes?

Is this method suitable for multiple plumes in a single field? I'm guessing that it might be, but it would be great to have that explicitly stated one way or another and what procedure would have to be followed to allow fluxes with overlapping plumes be separated (or joined) to give an overall flux.

We assume the system would sense multiple plumes if they were sufficiently spatially distinct, but clearly overlapping plumes would require a more complex analysis. The issues would be similar to mass balance experiments using aircraft. We have added a comment regarding this issue to the modified conclusion (see below).

1  General details on equipment used.

Can all items of equipment used for measurement or quantification be defined throughout the paper, there are mass flow controllers, anemometers etc without provenance.

Yes, we have added that information in the new appendix.

1  Explanation of the rationale behind the controlled release set up and fluxes used.

Ideally the controlled release would have had a number of releases at different rates under different wind conditions. This really only gives us an understanding of the capability of the set up under this specific set of conditions. I would like to understand the reasoning for only testing under such limited scenario and would recommend the authors to consider taking part in something along the lines of Adam Brandt's group tests of instruments at the next possible opportunity (e.g. https://doi.org/10.1525/elementa.2022.00080).

We certainly agree that it would be ideal to have tested the system under a much wider range of releases and wind conditions. The limitations were imposed by the available resources as this system was not developed as part of a larger program that could provide access to, for example, the Methane Emission Technology Evaluation Center at Colorado State University used by Bell et al. (The suggested reference that was also in the original manuscript.) The eventual goal of this system would be the detection of multiple gases through DCS rather than a single gas. Therefore, further tests would also want to incorporate multiple gases at multiple release levels and wind conditions. We have added a comment to this effect in the modified conclusion.

Minor

Fig 2. Can the averaging time be expanded so that the minimum in the Allen variance can be seen in the $CH_4$ measurement precision.

The figure has been modified. Interestingly, there is no minimum out to a few minutes. We were not able to further extend the Allen variance due to the limited UAV flight time. The new figure is:

[Figure]

Note that it starts at 4 s now instead of 1.6 s because we used a different flight that lasted longer. However, since sensitivity is lower for this flight overall than the previous flight, we changed the sensitivity numbers in the paper accordingly.

L121. Requires explanation as to why this emission rate and location is representative of a real emission.

We have added the sentence regarding the emission rates:

*These flow rates correspond to a leak of ~0.7 kg/hr for methane, corresponding to a practical lower bound for systems detecting methane leaks in oil/gas fields (Ravikumar et al., 2019; Johnson et al., 2021; Bell et al., 2022). The flow rate for acetylene was chosen to provide a similar signal level; future tests with multiple species would be coupled with the required species-dependent sensitivity.*

The conclusions feel rather light, it would be good to understand where the group feels the strength of this method is compared to the other technologies on the market.

We have expanded the discussion in the introduction slightly (and added additional reference of Johnson et al. 2021) to add more context as:

*Finally, several mass-balance approaches have been demonstrated using column-integrated measurements including solar-occultation flux (which can only be used during daytime/sunny conditions) (Mellqvist et al., 2010; Kille et al., 2017) and airborne LiDAR (which has focused on methane or carbon dioxide) (Ravikumar et al., 2019; Amediek et al., 2017; Bell et al., 2022; Kunkel et al., 2023; Johnson et al., 2021). There are two significant distinctions between these LiDAR approaches and the approach discussed here. First, the LiDAR systems are mounted on a larger aircraft, which has added cost and complications but does not require a van and can more easily cover a large area. Second, the LiDAR targets a single species, which is well suited to finding methane leaks in an oil/gas field, for example, while the system here relies on broadband dual-comb spectroscopy that can detect multiple species. If used in conjunction with a mid-infrared dual-comb system, this approach could then simultaneously detect multiple volatile organic compounds beyond methane.*

In addition, we have modified the conclusion to address this and several other points raised by the reviewers:

*This study was designed as a proof of concept for the method. Due to available resources and other logistical considerations, as well as some equipment malfunctions, the measurements were limited in scope. The next step is to do more extensive testing over a range of release conditions, for example, at a facility such as the Methane Emissions Technology Evaluation Center (METEC) (Edie et al., 2020; Riddick et al., 2022).*

*This new methodology has several potential advantages compared to other flux measurement methods. First, a key advantage is that no atmospheric dispersion model is needed since the flux is determined directly from the data. This also means that multiple spatially separated sources or areal sources can also be measured, although in an area with many sources, care needs to taken to make sure that the background is determined properly to just detect the sources of interest. A limitation of the specific flight pattern shown is that the source location needs to be approximately known; however, this limitation can be overcome either with a priori information (e.g., if the equipment or facility to be measured is known) or by performing spatial scans first. In addition, modified flight patterns such as flying vertically and then horizontally toward the source could likely overcome this limitation. Second, the methodology is flexible, so it can be used to determine fluxes for any gas that can be measured with open-path dual-comb spectroscopy or other open-path spectroscopy such as active differential optical absorption spectroscopy (Stutz et al., 2016). In particular, $CO_2$, $NH_3$, HDO, ethane, formaldehyde (HCHO), CO, and $N_2O$ have all been measured with DCS (Waxman et al., 2017; Ycas et al., 2019; Giorgetta et al., 2021; Herman et al., 2021). By only requiring the lightweight retroreflector to be flown, a small UAV can be used regardless of the gas or gases to be measured.*

*Combined, these advantages give the capability for rapid, easily deployable, multispecies flux measurements from point or distributed sources. This could be beneficial for example to survey emissions from fields, agricultural facilities, wastewater treatment plants, and oil and gas facilities. In addition, with further engineering of mobile DCS, measurements could be conducted from a moving van (similar to the solar occultation flux technique), allowing for flexible and rapid coverage of a wide area.*